# Water–Sulfuric Acid Foam as a Possible Habitat for Hypothetical Microbial Community in the Cloud Layer of Venus

**DOI:** 10.3390/life11101034

**Published:** 2021-09-30

**Authors:** Dmitry A. Skladnev, Sergei P. Karlov, Yuliya Y. Khrunyk, Oleg R. Kotsyurbenko

**Affiliations:** 1Research Center of Biotechnology of the Russian Academy of Sciences, Winogradsky Institute of Microbiology, 119071 Moscow, Russia; skladda@gmail.com; 2Network of Researchers on the Chemical Evolution of Life, Leeds LS7 3RB, UK; 3Department of Urban Studies, Moscow Polytechnic University, 107023 Moscow, Russia; ekems@mospolytech.ru; 4Department of Heat Treatment and Physics of Metal, Ural Federal University, 620002 Ekaterinburg, Russia; iu.ia.khrunyk@urfu.ru; 5M.N. Mikheev Institute of Metal Physics of the Ural Branch of the Russian Academy of Sciences, 620108 Ekaterinburg, Russia; 6High Ecology School, Yugra State University, 628011 Khanty-Mansiysk, Russia

**Keywords:** Venusian clouds, water–sulfuric acid foam, microbial community, extremophilic microorganisms, aero-geochemistry

## Abstract

The data available at the moment suggest that ancient Venus was covered by extensive bodies of water which could harbor life. Later, however, the drastic overheating of the planet made the surface of Venus uninhabitable for Earth-type life forms. Nevertheless, hypothetical Venusian organisms could have gradually adapted to conditions within the cloud layer of Venus—the only niche containing liquid water where the Earth-type extremophiles could survive. Here we hypothesize that the unified internal volume of a microbial community habitat is represented by the heterophase liquid-gas foam structure of Venusian clouds. Such unity of internal space within foam water volume facilitates microbial cells movements and trophic interactions between microorganisms that creates favorable conditions for the effective development of a true microbial community. The stabilization of a foam heterophase structure can be provided by various surfactants including those synthesized by living cells and products released during cell lysis. Such a foam system could harbor a microbial community of different species of (poly)extremophilic microorganisms that are capable of photo- and chemosynthesis and may be closely integrated into aero-geochemical processes including the processes of high-temperature polymer synthesis on the planet’s surface. Different complex nanostructures transferred to the cloud layers by convection flows could further contribute to the stabilization of heterophase liquid-gas foam structure and participate in chemical and photochemical reactions, thus supporting ecosystem stability.

## 1. Introduction

Owing to new data on Venus obtained during space missions and concepts which suggest the existence of life on this planet in the past and/or present, the astrobiology of Venus has been rapidly developing in the last few years. According to the data provided by the Venus Express spacecraft, the upper atmospheric layers of Venus are constantly losing water molecules [1,2,3,4,5]. The extrapolation of this process gave a birth to hypotheses on the existence of a larger amount of water, and even an ocean during ancient times when the conditions on Venus were milder and more habitable for Earth-type living organisms [6,7,8,9]. The idea for the existence of microbial life in Venusian clouds was first proposed after getting data on the physical and chemical parameters there [10]. This hypothesis was further discussed and developed [11,12]. The discovery of an unknown compound within Venusian clouds, the UV absorption spectrum of which was similar to that of biomacromolecules, has significantly supported this habitability concept [13,14]. Recently, all the ideas related to hypothetical Venusian microorganisms and their possible biochemical properties were summarized [5,15,16,17,18]. Astrobiologists consider Venusian clouds to be a habitable environment for extreme forms of life similar to extremophiles on Earth based on a habitability concept that includes: (1) the presence of a solvent needed for biochemical reactions, (2) appropriate physicochemical conditions, (3) available energy sources, and (4) biologically relevant elements C N P S O H [8,18]. Finally, Seager and co-authors suggested a scheme describing the life cycle of microbial cells within Venusian clouds which would enable the preservation of the critical microbial biomass needed for the reproduction and functioning of the whole microbial system [4]. In this study, we discuss the main conditions providing a sustainable ecosystem of putative Venusian microorganisms and suggest an original hypothesis on a heterophase water-gas foam structure habitat where various trophically linked microorganisms could exist as a microbial system the most efficiently.

### 1.1. How Could a Microbial Community Have Been Formed in the Clouds of Venus during Its Catastrophic Overheating?

Most likely, with the increase of surface temperatures on Venus, a direct selection for (poly)extremophilic forms of microorganisms occurred. Indeed, thermophilic and halophilic Archaea and Bacteria can thrive on Earth in similar conditions [19,20,21,22]. The drying of water bodies at the significant elevation of temperature undoubtedly was accompanied by a strong increase in salt concentrations. According to the data obtained, the clouds of Venus contain up to 75% sulfuric acid [2,3,4,17]. Under such extreme conditions, a strong selection for extreme acidophilic organisms might have occurred [23,24,25,26]. Prior to catastrophic climate change on Venus, all the types of extremophilic microorganisms would have occupied the appropriate ecological niches. Further, the zone of “ecological comfort” for extremophiles had been expanding. The rise of temperature on the planet was the most advantageous for thermophilic forms. When the temperature increased above 100 °C, the entire volume of water evaporated and was transferred into the cloud layer together with water-soluble organic and inorganic compounds. With the same ascending streams of hot water vapor, both active viable cells of microorganisms and spores could have been transferred to the clouds (Figure 1).

The entire evaporation of water from the planet’s surface must have been accompanied by the formation of abundant cloud masses at the altitudes where water vapor could have cooled down and condensed. Most probably, the pH of formed clouds was initially close to neutral, hence various microorganisms (not necessarily acidophiles) could have thrived there. Only later, after Venus’s atmosphere had lost the main amount of water, the adaptive selection of organisms was directed towards a resistance to a highly acidic environment. Following the complete evaporation of water from the planet’s surface, further heating of Venus (from 100 °C to over 400 °C) could have caused dust storms, thermal decomposition of organic and inorganic materials, and the active transition of dehydrated products into the atmosphere by upward convection gas flows. Additionally, in the beginning of catastrophic overheating, the products of decomposition of dead biomass on the planet’s surface could have been transferred to the water phase within clouds with water vapors, and hence became the source of different organic nutrient compounds for the development of a newly formed community of microorganisms. The microorganisms have been further adapting to the constant increase in the acidification of the cloud layer gradually losing water [1,10,16].

To survive in unfavorable changing conditions, a hypothetical microbial community in the Venusian cloud layer could have occupied a specific ecological niche represented by the liquid phase of a hypothetical unified water volume where dissolved nutrients necessary for cells are transported most efficiently. Unlike aerosols, such a foam structure allows the spread of cells increasing in number over the whole space of the ecological niche. Indeed, without the unified volume of water it is difficult to imagine the effective trophic interactions between different cells as well as the transport of organic and inorganic metabolites and other compounds (nutrients and inhibitors) which could influence the development of microorganisms. In addition to the concept of the origin of life on the surface of ancient Venus, we also consider a possibility that life forms were delivered to the planet via panspermia (interstellar lithopanspermia or ballistic panspermia within the Solar system) which could have occurred at any historical stage of Venus (Figure 1).

### 1.2. The Hypothesis of Water–Sulfuric Acid Foam in Venusian Clouds as a New Type of Ecological Niche

The proposed hypothesis is based on a clear understanding that the habitat necessary for a stable functioning of microbial communities should contain a unified water volume providing for the movement of individual cells and, more importantly, the free exchange of organics and inorganics dissolved in the medium which enables the possibility of metabolic cycles of biologically important elements. We assume that the conditions in both ancient and modern Venusian clouds are favorable for the formation and existence of foam structure, i.e., favor the closing of separate water formations resulting in hollow spheres or bubbles filled with gas, stabilized by surfactants and surrounded by Plateau borders and nodes (Figure 2). Indeed, a branched network of channels filled in with a foam water phase forms a unified water volume along the foamiest part of the structure which is accessible to all the cells residing there. This unified network of all Plateau channels can facilitate the proportional spreading out of all soluble compounds across the whole liquid volume.

The molecules of dissolved compounds able to form hydrate bonds with water molecules increase the surface tension force and thereby essentially stabilize the foam structure [27]. A high metabolic activity of cells contributes to the further formation and stabilization of foam, particularly due to the ability of microorganisms to produce biosurfactants [28,29]. The possibility of its existence and stabilization is based on the well-known physical–chemical aspects of the formation of water–gas heterostructures [30,31,32,33,34]. Moreover, in water microbial ecosystems on Earth, the natural processes of water foam formation are well recorded [35,36,37,38]. Importantly, in contrast to the previous ideas on the existence of microorganisms within aerosols, our hypothesis of a habitable foam overcomes the problem of an isolated existence of microbial cells in a small volume of droplets within clouds. Indeed, the latter is hardly suitable for the existence of a true balanced community of microorganisms.

### 1.3. The Main Physical-Chemical Properties of Liquid-Gas Foams

The foam is a complex system with a gaseous dispersed phase and a liquid/solid dispersed medium. Most often, natural foam consists of gas bubbles surrounded by water films. Such a structure is considered to be a polydisperse non-equilibrium system as gas bubbles can be of a different size. Should the liquid film break, the whole liquid is immediately absorbed by a Plateau channel (Figure 2). According to Plateau laws, the edges of the foam bubble are channels filled in with dispersed medium. In one channel, thin liquid films meet in threes at the angle of 120° [39].

The main parameters providing foam stability are surface tension and the Gibbs’ effect of the surface elasticity of thin liquid films which are reflected in the stability of separate bubbles [40,41]. Surface active substances (surfactants), while being added to the water phase, reduce surface tension at the gas-liquid boundary and decrease energy loss which results in the stabilization of the foam structure. A decrease in the pressure difference between neighboring bubbles slows down the transferring of gas from small pores (bubbles) into the large ones. Bubbles of anisotropic shape and/or large size are characterized by higher excessive surface energy; hence they tend to break down more easily. Foam dynamics is regulated by three main processes: growth, drainage and breaking down. With time the amount of foam bubbles decreases while their size increases which is caused by different pressure between neighboring bubbles. The drainage is the process of thinning the films, i.e., bubble walls, due to water flowing off under gravitation.

Low acidity should not affect the stability of foam structure as pH values do not influence surface tension of water solutions. Foam stability depends much more on the concentration of dissolved substances, pressure, temperature, viscosity of solution, diffusion coefficient of the foaming agent, the amount of agent in the total volume, specific internal energy of system components, etc. [41].

The temperature directly affects the stability and structure of foams. Indeed, in the middle cloud layers of Venus, at an altitude of 60–65 km, the temperature is close to 0 °C, which might cause the condensation of foam. This layer of foam may contain water droplets and ice crystals. At lower altitudes, however, the volume of foam bubbles may expand with the reduction of medium viscosity. A quite low concentration of water vapor, 0.003%, is a complicated factor in relation to the existence of foam structure. Notably, the foam can easily change its structure, shape, and volume. Thus, contributing to the establishment of contacts between microbial cells [36]. Additionally, the movement of foam masses within the whole cloud layer facilitates the transfer of microorganisms to different habitable zones with new environmental conditions within Venusian clouds.

### 1.4. Formation and Stabilization of Water-Sulfuric Acid Foam with Different Additives

All the above-mentioned data on the formation and stabilization of foam, obviously, require experimental verification by modeling conditions resembling those of Venusian clouds. We have obtained foam structures with highly concentrated sulfuric acid and used various additives to study its stability. The short description of the experiment and its main results are presented below.

Briefly, dissolved additives were introduced into tubes (250 mL) and then liquid phase was added (50 mL). As biogenic additives cell lysates of Archaea (*Halobacterium salinarum*) and eucaryotes (yeast extract) were tested. Tubes were closed with stoppers and shaken intensively for 30 s to form foam. The thickness of foam layer was measured after 15 s. The stability of foam layer was analyzed after 15 min.

The pure water, as well as both solutions of sulfuric acid (50% and 75%), did not form stable foam (Table 1). The presence of additives, however, led to the formation of well-structured and stable foams during and after intensive shaking of solutions. The appearance of large bubbles observed at 15 min following shaking indicate the stability of foam layers (Figure 3 and Figure 4). Water foams with cellular lysates showed a large thickness (up to 20 mm). With polyvinyl alcohol added, a thinner layer of water foam was formed and even less structured layers were observed in the presence of sulfuric acid. The effect of additives was lower in the most concentrated sulfuric acid solution (except for colloidal iron FeCl**_3_**). The most pronounced sulfuric acid foam was registered with *H. salinarum* lysate as an additive

It should be noted that all types of foam showed a high stability, i.e., their thickness did not decrease by more than two or three times in 15 min (except sulfuric acid foam with polyvinyl alcohol that was halved in 3 min). It was noted that in foam types with a high content of sulfuric acid the temperature of solutions in the range from 30 °C to 80 °C did not affect the stability of the formed foam.

Thus, this experiment indicates that the organic compounds of microbial cells act as stabilizers of foam formed mechanically from highly concentrated sulfuric acid. More studies are needed to understand what kind of biogenic and inorganic compounds can act most effectively as stabilizers of such water–sulfuric acid foam.

### 1.5. The Basic Mechanisms of Foam Formation in Earth Ecosystems

On Earth, stable natural foam structures are based on specific compounds present in the water phase enhancing surface tension and stabilizing the foam [42,43,44]. A good example of the generation of stable water foams is the boiling of milk. Protein and fat molecules stabilize bubbles formed from air dissolved in milk. Such foam is rather stable even when the temperature is decreased. The effects of many dissolved compounds on foam formation and its stability have been recorded in Earth conditions [28,45]. We could speculate that foam structure within Venusian clouds form under analogous conditions, especially if stabilized by biosurfactants, is governed by the same principles.

The heterophase system (water droplets in air phase) of Earth’s atmosphere has been well studied, supramolecular forces acting at the boundaries of phases, conditions affecting the size and stability of droplets etc. have been extensively analyzed [39,40,45]. It is reasonable to assume that the laws behind intermolecular relations, the forces of intermolecular attraction and equilibrium distances between the surface layer molecules in liquid-gas systems on Venus are analogous to those on Earth.

It was shown that water droplets are formed from large water volumes only if they are mechanically shed off, i.e., when water is ripped off a wave crest by the wind or originate from waterfalls (Figure 5B) [46]. In clouds, cooled down water vapors condense into drops (the formation of liquid phase within the gas phase). These drops can exist and move in the atmosphere for some time, depending on physical conditions [36,37]. When a gas bubble leaves its water environment and moves to the gas phase, a second external boundary layer of water molecules is formed. Hence, the surface of gas bubbles within the gas phase is represented by a three-layered structure (three phase system) with two surface layers of water molecules, where complicated supramolecular relations at the phase boundaries occur (Figure 5B,C). The development of a foam structure begins from the formation of gas clusters as embryos of gas bubbles inside the water environment. The molecules of dissolved gases are capable of forming clusters, i.e., molecular aggregations which basically represent a new phase inside water (a new gas phase in a liquid phase) (Figure 5A). Numerous studies obtained accurate data on fine mechanisms of interactions between the boundary layers of water molecules (forming the surface of gas bubbles in water phase) with the molecules of the deeper part of water phase [35,39]. Due to the strong interactions between closely located molecules local (non-stable) ordered groups containing several molecules can be formed. This phenomenon is known as short-range order.

It is known that gases dissolved in water are present in it in the form of so-called charged nanobubbles with a rather high density. In the first quarter of the last century, it was shown that electrical charges are present on the surface of an air bubble in water [47]. Their origin was explained by selective (depending on charge sign) surface adsorption of ions that are always present in water. An alternative term, bubston (bubble, stabilized by ions) was suggested [48] and the dependence of bubston stability from solution properties was determined [49,50]. We suggest that nanobubbles, bubstons, are the basis for the formation of larger gas bubbles in water droplets present in Venusian clouds.

The aforementioned studies revealed that gas bubbles in water phase are characterized by a high mobility and plasticity, i.e., they can change their shape. Notably, the layers of water molecules at the gas-water boundary of phases can transform bubstons into larger gas clusters. Additionally, the boundary layers do not hinder the interaction of neighboring clusters which results in a gradual growth of gas bubbles (Figure 5A). Such abiogenic process of gas bubble growth has an irreversible character.

Another explanation for the origin of foam formation in Venusian cloud layers is the boiling of separate droplets located in the lower layers due to upward heated flows of atmosphere. The heating of droplets within the water phase being in equilibrium with the external gas environment (for example, atmospheric air) can cause the development of larger gas bubbles (Figure 5C). With a high probability, liquid being close to the boiling point or in an overheated state gives the birth to vapor bubbles, precursors of a new phase, which can lead to the formation of a separate extra-droplet bubble.

Alternatively, the formation of three phase three-layered (gas–liquid–gas) bubbles is possible in the presence of metabolically active microbial cells capable of releasing gaseous products into the foam gas phase. Such phenomenon is quite common for the Earth-type microorganisms. We can assume that gas, released by microorganisms inhabiting Venusian clouds, is collected in the liquid phase (for example, in aerosol droplets) as separate bubbles which would further become larger (Figure 5D). Moreover, following a certain period of time, after cell growth and death water phase might contain biogenic organics increasing the environmental density. We should note that the release of gaseous metabolites by microorganisms can occur inside separate droplets of water aerosols. With the growth of a bubble, the medium should be equally pressed to the surface of the droplet. The presence of interacting microorganisms in clouds allows us to assume that the content of bubble gas phase would serve as nutrients for other organisms resulting in the accumulation of both biomass and its decomposition products within the droplet. It is possible that the stabilization of droplet surface tension might lead to the charge removal from droplet surface, thus preventing the repulsion of droplets and allowing them “to touch”. The pressure from the contact, in its turn, can cause the formation of pores in surface layers (menisci) between droplets which could result in the liquid phases merging at local points of Plateau borders. Hence, aerosol droplets can be transformed in the foam-like structure with cells inside it (habitable foam) (Figure 5E).

### 1.6. The Biogenic and Abiogenic Stabilizers of Foam Structure

In order for the formed foam to exist in a cloud layer, the heterophase system requires the presence of surface-active surfactants or stabilizers at phase boundaries. In the Earth atmosphere, clouds are formed from condensed water vapor almost without surfactants. On the contrary, we assume that the water phase of Venusian clouds contains surfactants of different chemical nature which could have been transferred with water vapor earlier and currently they appear due to exchange of matter. Notably, a variety of studies have been dedicated to the problems of microbial-induced foam formation. The representatives of *Actinobacteria* and other genera (*Clostridium* XI, *Arcobacter*, *Flavobacterium*, *Gordonia*) can be found in natural foams and are known as biosurfactants producers [36,38,51]. Surfactant molecules are placed predominantly on the surface of air bubbles stabilizing Plateau borders and nodes and forming a stable film which increases the resistance of foam bubbles to coalescence [52].

Microbial surface-active compounds are classified based on their molecular structure. High molecular weight compounds include lipoproteins, lipopolysaccharides, heteropolysaccharides, proteins, polymeric, and particulate compounds. Low molecular weight compounds consist of glycolipids (rhamnolipids, sophorolipids, mannosylerythritol lipids, and trehaloselipids) and lipopeptides (e.g., surfactin, fengycin) [22,38]. The metabolic function of biosurfactants in bacteria is to allow the cells to use water-insoluble substrates by acting as natural emulsifiers. Biosurfactants are usually characterized by low toxicity, biodegradability, increased environmental compatibility and good stability at a wide range of pH and temperature. Most polysaccharides lack an amphiphilic structure and cannot adsorb to fluid interfaces. The stabilizing ability of polysaccharides is associated with their ability to thicken solutions which slows down the destabilization of foams. Another common feature of polysaccharides is their function as cryoprotectants [36].

We can also suppose that specific organisms which emerged in the course of the biological evolution of Venusian microbial communities could synthesize unknown biopolymers or membrane compounds which form special stabilizing monolayers at the foam phase boundaries.

Another large group of inclusions which could stabilize the hypothetical habitable foam of Venusian clouds can be composed of abiogenic compounds such as metal particles, highly dispersed mineral powders, silica particles, and other inorganic compounds or crystal structures [35,42,51,53,54]. Moreover, the examples of inorganic foams formed from materials common on Earth and under temperatures typical for Venus are known. For instance, a porous xerogel is formed by sol–gel process at subcritical drying of gel employing temporary pore fillers or solid framework support (e.g., carbon or organic) that are removed by thermal oxidation. Such abiogenic inorganic stabilizers of foam can be light and volatile. Thus, winds and upward flows of heated gases can lift them to the clouds at high altitude [55,56,57].

### 1.7. The Synthesis of Complex Organic Compounds on Venus’s Surface and Their Possible Contribution to the Stabilization of a Hypothetical Habitable Foam in the Venusian Cloud Layer

Extremely high temperature and pressure on Venus’s surface exclude the possibility of Earth-type life on its surface [24]. Nevertheless, such conditions allow the chemical reactions resulting in the abiogenic synthesis of complex polymers [54,58]. Such polymerization reactions occur in several stages. The oxides of different metals (FeO, SiO**_2_**, Al**_2_**O**_3_**, MgO, CaO, K**_2_**O) widely distributed in Venus’s ground may serve as catalysts for chemical reactions with atmospheric gaseous compounds (N**_2_**, CO**_2_**, H**_2_**, H**_2_**S) resulting in the generation of iron nitrides (Fe**^x^**N) or complexes Fe**^x^**NH and Fe**^x^**NH**_2_**, as well as FeS. These compounds can further participate in synthesis reactions of various organic compounds, i.e., CH**_3_**SH, COS, CH**_2_**O (Figure 6).

In the presence of an iron catalyst and highly accessible atmospheric CO**_2_** and N**_2_**, these compounds can be further involved in the synthesis of high molecular compounds of g-C**_3_**N**_4_** formula, due to their graphite structure and triazine monomers. According to the hypothesis of Ksanfomality [58], such types of complex compounds could have been the basis for the development of life forms that are able to persist on Venus’s surface. In the framework of our hypothesis we are interested rather in possible participation of such thermoresistant organic polymers in the exchange of matter with the cloud layer resulting from strong convection flows. Such compounds can play a few roles in the cloud layer system. Firstly, they can serve as a nanomatrix for the catalysis of chemical denitrification yielding the generation of gaseous products from nitrite [59,60]. Secondly, they can contribute to the stabilization of foam structure [39,51]. Finally, being raised to higher altitudes such compounds may decompose during photocatalytic reactions [61] and serve as a transporter and an additional carbon source for the cloud microbial community (Figure 1).

### 1.8. The Basic Conditions for the Stable Existence of Microbial Community

The habitability of terrestrial microbial ecosystems is defined by the presence of microbial communities involved in balanced trophic relationships and cyclic exchange processes of biologically important elements [62,63]. The higher diversity of metabolic pathways occurring at the decomposition of organic matter within the system, and the higher adaptive properties of microbial communities under the changing environment [64,65]. An important condition for all the above-mentioned adaptation strategies to drastically changing environmental parameters is the presence of a unified water volume in the habitat where all the necessary metabolic exchange reactions and cycling biological processes occur [18,62,63].

The stability of ecological niches is based on the most important principle of the recovery of reaction components involved in energy release reactions driven by microorganisms. We should note that the stability of microbial communities does not contradict their ability to contribute to the stabilization of foam structure, in particular, owing to their ability to produce gaseous metabolites and to the synthesis of organic compounds acting as surfactants [40,51]. Such a hypothetical microbial community inhabiting the Venusian cloud layer can consist of thermo-acidophilic chemolithotrophic anaerobic microorganisms (Figure 1). They consume energy released in oxidative-reduction reactions involving sulfur and iron [18,64]. Microbial sulfate and iron reduction coupled with iron-dependent denitrification which can result in the formation of gaseous products is among the simplest biochemical cycles possibly occurring in this ecosystem (Figure 7). Such a system can function in the liquid phase of a foam structure where the microorganisms actively interact with solid particles carrying different elements, mainly sulfur and iron.

Simultaneously, microorganisms can release different stress regulators in order to withstand extreme physical–chemical factors of the environment. These substances, in addition to biogenic organic matter, can function as surfactants and contribute to the stabilization of the foam structure.

It is assumed that the Venusian cloud layer at the altitudes within the temperature range of 50‒80 °C are the most suitable zone providing a new type of extraterrestrial ecological niche with sulfuric acid-water foam structure as described above [14,18,57]. Presumably, at higher altitudes under increased exposure to UV and cosmic radiation, foam structure can actively participate in photocatalytic processes of organics and water decomposition [66].

If hypothetical Venusian microbial forms are similar to Earth-type living organisms, their cells are enclosed by membranes formed by polar lipid bilayers. In this case, fat water-insoluble compounds of decomposing cells are excluded from the circulation of organic carbon, as it occurs on Earth [62]. Lipids and fatty acids might gather at the surface of the water phase. In Earth conditions, over a billion years, this process led to the formation of oil. Within habitable Venusian clouds, water-insoluble fats (oil precursors) can be located on the surface of clouds (both bubble-like and foamy clouds) directed towards Space, from the planet’s gravity. Most likely, the orientation of polar molecules would lead to the formation of monolayers at the phase boundaries and, eventually, to the stabilization of foam.

## 2. Conclusions

The authors suggest a novel hypothesis on the existence of a specific ecological niche in the cloud layer of Venus supporting the hypothetical microbial community. This habitat is composed of water–sulfuric acid foam that forms a unified internal volume providing conditions for a stable existence of the microbial community as a complex biological system. Indeed, under conditions of unified extended space filled in with a liquid phase, trophic interactions and metabolic cycles of all biologically important elements, that are needed for a microbial community to thrive, can occur.

We assume that such foam structure within Venusian cloud layer could have resulted from the evaporation of the whole water volume from Venus’s surface following its catastrophic overheating. Together with water vapor, active viable cells of microorganisms as well as diluted organic and inorganic compounds, which could serve as nutrients, could have been transferred to the clouds. The cloud layer was gradually losing H**_2_**O due to photodissociation. Simultaneously, the saturation of water vapor with sulfuric acid resulted in cloud acidification. Hence the microorganisms must have adapted to significant changes in the planet’s environment. In such conditions, water–concentrated sulfuric acid foam structure can preserve stability due to the presence of biogenic organic compounds and an additional intensive transfer of heteropolymeric nanocomplexes that could be synthesized at the planet’s surface under high temperatures. Additionally, various inorganic compounds such as FeCl_2_ could also act as surfactants for water–sulfuric acid foam. Moreover, gaseous products released by microorganisms can contribute to the enlargement of the foam volume. Finally, with mobile foam fragments microbial cells can be spread to all the habitable zones within Venusian cloud layers

Thus, we postulate that the physicochemical conditions in the cloud layer of Venus may be favorable for the formation of a stable heterophase foam structure capable of providing all the necessary trophic interactions for the stable existence of a microbial community. This hypothetical ecological system can be quite stable due to the presence of various biogenic and abiogenic surfactants.

The authors hope that the hypothesis presented in the article will inspire other research including empirical works which is quite important in the search for life on exoplanets.

## Figures and Tables

**Figure 1 life-11-01034-f001:**
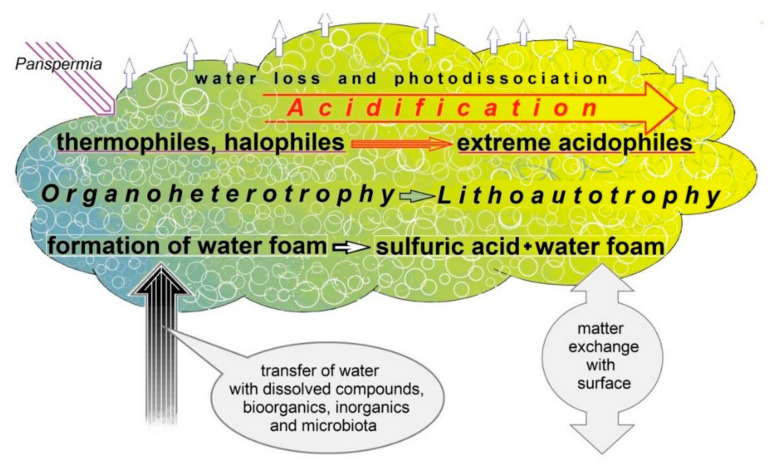
The scheme of possible key events that would lead to the formation of a new type of ecological niche for the habitat of microorganisms in the clouds of Venus.

**Figure 2 life-11-01034-f002:**
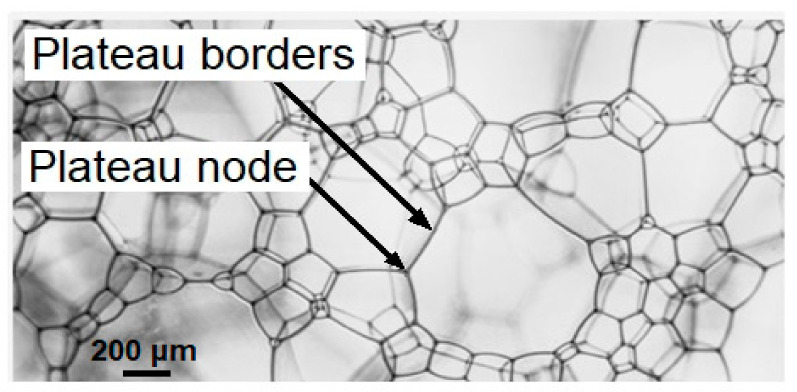
Specific water foam heterophase polydisperse structure suggested for the habitable ecosystem in the Venusian cloud layer.

**Figure 3 life-11-01034-f003:**
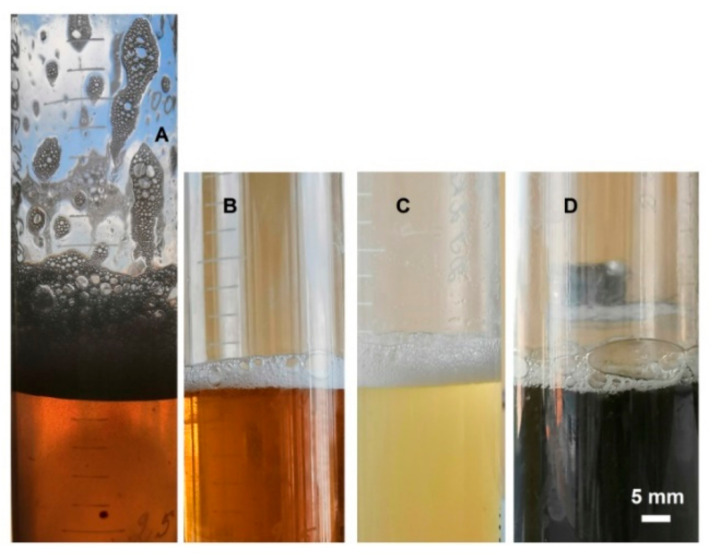
The foam layer observed at 15 min after shaking. (**A**,**C**)—containing water as liquid phase; (**B**,**D**)—with 50% H_2_SO_4_; (**A**,**B**)—with *H. salinarum* biomass lysate as additive; (**C**,**D**)—with yeast extract as additive.

**Figure 4 life-11-01034-f004:**
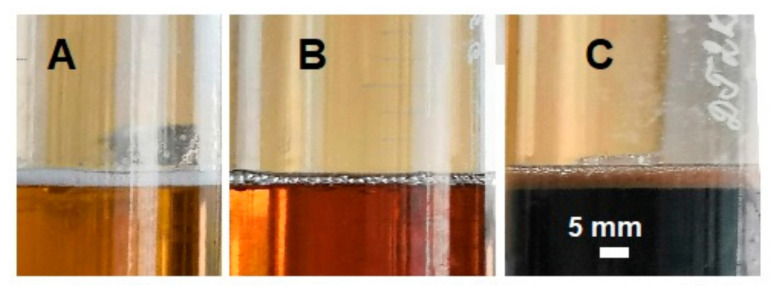
The foam layer with 75% H**_2_**SO**_4_** as liquid phase observed at 15 min after shaking. (**A**)—with colloidal iron FeCl**_3_** (1%) as an additive; (**B**)—with *H. salinarum* biomass as additive; (**C**)—with yeast extract as additive.

**Figure 5 life-11-01034-f005:**
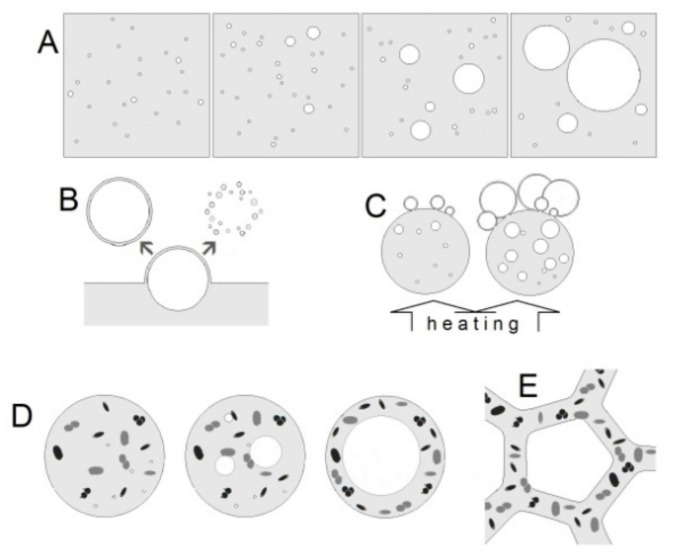
The scheme of gas bubbles formation in water phase from bubbstones (**A**); mechanically on water surface (**B**); in heated drops (**C**); by microbial gas generation (**D**); followed by foam formation (**E**).

**Figure 6 life-11-01034-f006:**
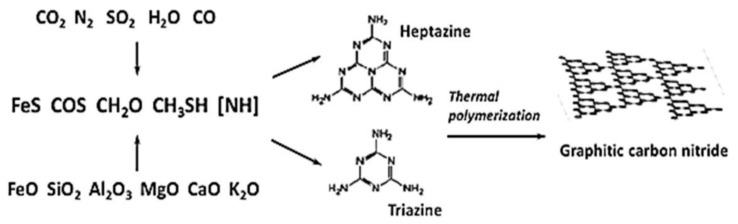
Possible chemical reactions of abiogenic polymerization at the surface of Venus.

**Figure 7 life-11-01034-f007:**
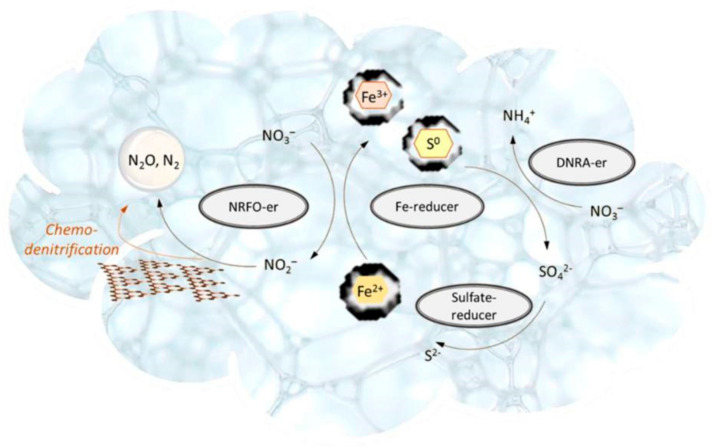
Possible biochemical processes occurring within foam continuous structure of Venusian clouds supporting a hypothetical microbial community.

**Table 1 life-11-01034-t001:** The foam formation in different media and the effect of different additives on its stability.

	Liquid Phase	H_2_O	H_2_SO_4_ 50%	H_2_SO_4_ 75%
Additive				
no additive		0	0	0
Archaea biomass **		18 *	12	8
eukaryotic cells *** (1% dry weight)		17	6	2
colloidal iron FeCl_3_ (1%)		0	0	4
polyvinyl alcohol (1%)		8	3	1

* The thickness of the foam layer (mm) observed at 15 s after the shaking was stopped. ** Freeze-dried *Halobacterium salinarum* biomass, at a concentration corresponding to the suspension of living cells, ~10**^6^** mL^−1^. *** Yeast extract (Difco, Lab., Detroit, MI, US).

## Data Availability

The study did not report any data.

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
