# Peer review of "Water–Sulfuric Acid Foam as a Possible Habitat for Hypothetical Microbial Community in the Cloud Layer of Venus"

_life, 2021, doi:10.3390/life11101034_

Round 1

Reviewer 1 Report

The revisions have significantly improved this manuscript and I feel most comments of the reviewers have been addressed.

However, I still feel there is a need to add that these foams do not occur in the earth's atmosphere and give a reason why.

The language remains over dense. The authors need to understand that for this journal they are writing for a biological readership and not a physical chemistry audience.

The language, while much improved, still needs further English language editing 

Author Response

Dear reviewer, we thank you for all your comments!

Below are our responses:

Rev. "However, I still feel there is a need to add that these foams do not occur in the earth's atmosphere and give a reason why."

We are going to add the explanation in the section "The biogenic and abiogenic stabilizers of foam structure":

"In order for the formed foam to exist in a cloud layer, the heterophase system requires the presence of surface active surfactants or stabilizers at phase boundaries. In the Earth atmosphere, clouds are formed from condensed water vapor almost without surfactants. On the contrary, we assume that the water phase of Venusian clouds contains surfactants of different chemical nature which could have been transferred with water vapor earlier and currently they appear due to exchange of matter. Notably, a variety of studies have been dedicated to the problems of microbial-induced foam formation".

We are waiting now for the reply of the reviewer 3 in order to make final corrections and upload the new version.

Rev. "The language remains over dense. The authors need to understand that for this journal they are writing for a biological readership and not a physical chemistry audience".

Our hypothesis is based originally on physico-chemical constraints and properties of the hypotetical environment so that we decided to present proper explanations in order to avoid additional questions. We believe it is important to give details for first presenting this concept. If you still think that some sections should be reduced, please specify them.

Rev. "The language, while much improved, still needs further English language editing"

We are going to do the final English editing after working with comments of the reviewer 3. Once this is done we will upload the final version of the manuscript.

Reviewer 2 Report

Thank you, the article can be accepted in present form.

Author Response

We thank the reviewer for valuable comments and suggestions!

This manuscript is a resubmission of an earlier submission. The following is a list of the peer review reports and author responses from that submission.

Round 1

Reviewer 1 Report

I enjoyed reading the manuscript authored by Skladnev at al.

Major comments:

-„A stable existence of microorganisms in the Venusian cloud layer“ would require a continues supply of nutrients, CHNOPS and other traces elements, for instance enzymatic co-factors such as Mo, V, etc. Not to mention a continuous supply of bioavailable phosphorus. It would be necessary that the authors provide an updated status of information on that (as a paragraph in their introduction or within a separated sub-chapter). Heterotrophism based on remnants of died cellular material is not sufficient on a long term to support proposed aerial microbiota in Venusian clouds, it may be helpful only for a certain short period after death of non-survivors.

-Help of English native speaker is absolutely needed for this manuscript.

-The authors need to provide an extensive prove of sulfuric acid-water foam structures. Under which specific conditions and with which specific compounds sulfuric acid would form foam structures? And how relevant these specific conditions are towards Venus’ clouds?

-Literally every second sentence of the manuscript requires missing references.

Minor comments:

L 46 "beings" – should be "living entities".

L 54 "aboriginal" – please rephrase

L53-56 any assumptions regarding pH values or pH extreme variations before, during and after climate change and how that would challenge hypothetical Venus’ cloud microbiota?

L62 modern biochemistry? Perhaps „biochemistry of Earth life forms” would be more suitable here.

L63-65 – need several references here.

L126-136 – every sentence needs several references here.

L189 and everywhere else: would be great to rephrase „foam-bubble structures“ with a more scientific terminology. How about monodispersed or heterodispersed or polydisperse foam, water-gas foam structures or any other scientific term?

L206-212: it needs a precise conclusion here: is it a foam structure which facilitates microbial aerial transport or microbiota who forms and stabilizes foam structure, or are there mutually beneficial/reciprocal relations in between those two?

L255-258 need references here. I doubt that low acidity and high saturation with sulfuric acid in Venus’ low could deck would not influence the stability of proposed foam structures.

L271: should be droplets here, not drops.

LL444-453 every sentence needs references.

L479: „Venusiam microorganisms“ – please rephrase, e.g., hypothetical microbial aerial life forms in Venusian clouds, etc.

Fig. 2 – please introduce scale bar here.

Reviewer 2 Report

In this manuscript by Skladnev et al., the authors extensively present the hypothetical scenario for the emergence, survival and succession of microbial communities in the early Venusian surface and afterwards in the Venusian atmosphere with the aid of water foam. 
In my view, this is a very interesting article, but major improvements need to be made prior to its publication. 
The language is understandable, but the paper needs to be reviewed by a native speaker for grammatical edits, that are important, since it is a hypothetical paper, sometimes it is not clear -due to grammar- whether the authors are referred to past studies or to their hypothesis. 
Also, the manuscript is not adequately referenced. I have made notes in the attached file with the lines where references are missing. 
Finally but most importantly, I raise certain issues regarding the microbial communities, all highlighted in the attached pdf file, that authors need to address, clarify, or elaborate before this paper is accepted. 

Reviewer 3 Report

The underlying premise of this manuscript, although very speculative, is novel and worthy of publication. I like the concept but feel the authors need to do more to make a convincing case that this scenario is feasible. They have demonstrates that foams could provide viable habitat for microbial life but I believe the authors also need to address the fundamental issue of whether foams can stably exist in any atmosphere. Clearly, they have never been detected in the earth's atmosphere and the authors do not convincingly demonstrate that they could exist in the atmosphere of Venus. This aspect need to more throughly addressed before the manuscript is suitable for publication. 

The manuscript should refocus solely on foam as habitat. The section 'How succession of biota could occur on Venus during its catastrophic overheating' should be removed. The suggestion of cloud foam as habitat is speculative enough without introducing the non relevant evolution and survival of life on the surface of Venus. The latter would be a suitable topic for a different manuscript.

The manuscript itself is poorly written, overly dense and difficult to follow. Some of this relates to the need to have an English native speaker thoroughly edit the language. The whole manuscript also needs to be restructured to improve the natural development of the ideas and also to simplify the expression. Sentence composition is unnecessarily complex.

In summary, I like the ideas expressed here and would like to see it published but if it is to have maximum impact, it must be improved